# Healthcare Workers’ Knowledge and Perception of the SARS-CoV-2 Omicron Variant: A Multinational Cross-Sectional Study

**DOI:** 10.3390/healthcare10030438

**Published:** 2022-02-25

**Authors:** Akshaya Srikanth Bhagavathula, Mohammadjavad Ashrafi Mahabadi, Wubshet Tesfaye, Kesavan Rajasekharan Nayar, Vijay Kumar Chattu

**Affiliations:** 1Institute of Public Health, College of Medicine and Health Sciences, United Arab Emirates University, Al Ain 17666, United Arab Emirates; 2Faculty of Medicine, Iran University of Medical Sciences, Tehran 14665-354, Iran; mjamahabadi@gmail.com; 3Sydney Pharmacy School, The University of Sydney, Sydney, NSW 2006, Australia; wubshet.tesfaye@sydney.edu.au; 4Global Institute of Public Health, Ananthapuri Hospitals and Research Institute, Thiruvananthapuram 695024, India; krnayar@gmail.com; 5School of Public Health, University of Alberta, Edmonton, AB T6G 1C9, Canada; vijay.chattu@mail.utoronto.ca; 6Center for Transdisciplinary Research, Saveetha Institute of Medical and Technological Sciences, Saveetha University, Chennai 600077, India; 7Department of Community Medicine, Faculty of Medicine, Datta Meghe Institute of Medical Sciences, Wardha 442107, India

**Keywords:** SARS-CoV-2, Omicron, COVID-19, awareness, knowledge, attitude, perceptions, prevention, survey, health professionals

## Abstract

In late November 2021, a new SARS-CoV-2 Variant of Concern (VOC) named Omicron (initially named B.1.1.529) was first detected in South Africa. The rapid spread of the SARS-CoV-2 Omicron variant became globally dominant, and the currently available COVID-19 vaccines showed less protection against this variant. This study aimed to investigate healthcare workers’ (HCWs) knowledge and perceptions about the novel SARS-CoV-2 Omicron variant. A cross-sectional anonymous electronic survey concerning the SARS-CoV-2 Omicron variant was conducted among HCWs during the second week of January 2022. The survey instrument was distributed through social media among HCWs to explore awareness (2 items), knowledge (10 items), source of information (1 item), and perceptions (10 items). Respondents who answered ≥80% of the items correctly were considered as having good knowledge and perception. A total of 940 of the 1054 HCW participants completed the survey (response rate: 89.1%); they had a mean age of 31.2 ± 11.2 years, most were doctors (45.7%), and most were from Asia (64.3%). All the participants were aware of the SARS-CoV-2 Omicron variant (100%). Only 36.3% attended lectures/discussions about Omicron and used news media to obtain information. Only a quarter of the HCWs demonstrated good knowledge (24.3%, *n* = 228) and perception (20.6%) about Omicron. However, while significant differences were observed in the knowledge and perception among HCWs, only a small proportion of doctors exhibited good knowledge (13%) and perception (10%) about the Omicron variant. HCWs who had participated in training/discussion related to the Omicron variant were more likely to have higher knowledge and perception scores (odds ratio: 1.80; 95% confidence interval: 1.04–3.11). As the SARS-CoV-2 Omicron variant spreads rapidly across the globe, ongoing educational interventions are warranted to improve knowledge and perceptions of HCWs.

## 1. Introduction

According to statistics from the John Hopkins University’s Coronavirus Resource Center, as of February 19, 2022, SARS-CoV-2 has infected over 421 million people globally, with over 5.8 million deaths from coronavirus in 2019 (COVID-19) [1]. In late November 2021, a new SARS-CoV-2 Variant of Concern (VOC) named “Omicron” (initially named B.1.1.529) was first detected in South Africa [2]. The rapid spread of the Omicron variant suggests that it may become globally dominant. In addition, the currently available COVID-19 vaccines have shown little to no protection against the SARS-CoV-2 Omicron variant [3].

The Omicron variant is less severe than earlier Variants of Concern (VOCs) [4,5]. Nevertheless, the rapid transmissibility of this variant continued to have a knock-on effect on services for other health conditions, leading to overwhelmed health systems across the globe. The risks of hospitalization, intensive care unit admission, and mortality, while considerably lower than were observed with the Delta variant [6], are still significant, largely affecting people who are not vaccinated [6]. Importantly, the existing vaccines provide the greatest protection against the Delta variant [7,8,9] and provide some level of protection against Omicron, especially in preventing severe symptoms that lead to hospitalization [10]. A longer time having elapsed from the first to the second dose was particularly associated with an increased risk of a symptomatic form of COVID-19 [11]. While more research and data are needed to fully understand this, evidence in certain settings suggests that the booster dose is effective in curbing the impact of Omicron [10,12,13].

Healthcare workers (HCWs), as frontline responders to the pandemic and members of a community highly affected by the VOC Omicron, have been key players in mitigating the effect of the COVID-19 pandemic and its consequences, as well as in implementing preventative measures suitable to evolving variants of interest. Several research articles have assessed various health specialties and populations’ knowledge, awareness, attitude, perceptions, and preventive practices towards COVID-19 [14,15,16,17,18]. However, given the evolving nature of the COVID-19 pandemic, updated knowledge of the epidemiological relevance of mutated versions of SARS-CoV-2 is imperative to understand the virulence and risk of transmissibility associated with emerging variants, as well as the effectiveness of existing public health and social measures. More importantly, assessing the knowledge of HCWs about the effective tools available to prevent newly emerging variants, the relevance of existing diagnostic tools in detecting such variants, and the efficacy of the range of vaccines currently in use is fundamental in reducing the multifaceted impact of the disease. As such, this survey aims to explore the knowledge and perception of HCWs concerning the SARS-CoV-2 Omicron variant to consolidate the evidence surrounding this new variant.

## 2. Materials and Methods

A cross-sectional web-based survey was conducted using Qualtrics^TM^ (www.qualtrics.com), and a survey instrument was distributed to obtain responses from HCWs globally during the second week of January 2022.

### 2.1. Content of the Survey Instrument

A 24-item survey instrument was developed using the information of the World Health Organization (WHO) and the US Centers for Disease Control and Prevention (CDC) on the SARS-CoV-2 VOC Omicron [19,20,21]. The survey was divided into four sections. The first section aimed to collect general demographic data such as age, gender, country, and profession. The second section explored the awareness (two items) and the personal sources of reliable information such as news, social media, health authority websites, and family/friends. The sources of information were scored using a four-point Likert scale (1—least used to 4—most used). The third section explored the understanding of SARS-CoV-2 VOC and their origins (five matching questions), specifically knowledge of the Omicron variant such as its origin (one item), symptoms (one item), transmissibility (one item), and risk prevention (one item), and the importance of COVID-19 vaccination (one item). The fourth section aimed to explore the perceptions of the SARS-CoV-2 Omicron variant and asked whether the respondents (yes/no/not sure): (1) believe the COVID-19 vaccine provides protection from the Omicron variant, (2) consider the COVID-19 vaccine booster dose important, and (3) know the influence of SARS-CoV-2 mutations on patient management. In addition, seven true or false questions were asked about the Omicron variant concerning vulnerability (one item), travel restrictions (two items), testing (one item), treatment (one item), protective measures (one item), and vaccination program (one item) (Supplemental File S1: survey questionnaire).

### 2.2. Validity of the Survey Tool

At first, the developed questionnaire and the materials used were distributed through email to five experts from different geographic regions to comprehensively assess the content domains of the questionnaire (using a scale of 1—poor to 5—excellent) and were encouraged to critically appraise the survey tool. The readability of the survey tools was assessed by distributing the questionnaire and materials to 15 randomly selected faculty members and asking them to rate the readability of each question from 0 to 100 (0–30: confusing; 31–50: difficult; 51–70: standard; 70–90: easy; and 90–100: very easy). Refinements were made, and the questions were reorganized to facilitate comprehension, following experts’ comments. Finally, the questionnaire was pilot tested with 15 randomly selected HCWs to assess clarity, relevance, and acceptability. These participants were not included in the research.

### 2.3. Sample Size

Sample size was calculated using the G*Power statistical software (version. 3.1.9.7; Heinrich-Heine-Universität Düsseldorf, Düsseldorf, Germany) [22]. The power analysis results indicated that an 80% power would be obtained with 95% confidence when 767 participants were included in the study (Supplemental: Appendix A). Considering 20–25% dropouts and participants not responding or giving incomplete information, we invited over one thousand individuals to participate in the survey.

### 2.4. Recruitment

The study questionnaire was distributed online to the study population through a Uniform Resource Locator Link of Qualtrics using social media sites such as WhatsApp, Facebook, LinkedIn, Telegram, Twitter, and personal emails of the HCWs. Furthermore, the survey link was advertised in health professional groups to reach the target population and was opened from 12 January to 15 January 2021. The survey tool was limited to one response of the questions per participant, and the survey instrument was made available for 30 min to read, comprehend, and answer all the questions.

### 2.5. Scoring

Each knowledge response was scored as “2” (correct) and “0” (wrong), with scores ranging from 0 to 20. Participants’ overall knowledge scores were categorized using a modified Bloom’s criteria cutoff point [23] as good if the score was between 80 and 100% (16–20), moderate if 60–79% (12–15), and poor if <60% (total score <12).

Perceptions towards the Omicron variant were assessed using ten questions, and the responses were graded as a 3-point Likert scale, an agreement scale ranging from ‘2’ for correct to ‘0’ neutral. The overall level of perception was categorized using modified Bloom’s criteria as good if the score was 80–100% (16–20 points), neutral if the score was 60–79% (12–15), and a misperception if the score was less than 60% (<12 points).

### 2.6. Statistical Analysis

The collected data were coded, validated, and analyzed using SPSS software version 24 (IBM Corporation, Armonk, New York, NY, USA). Descriptive statistics were used to summarize the information and presented as mean ± standard deviation or 95% confidence intervals (CI), median and range, and calculated frequencies and proportions. ANOVA test was used to compare the participants’ knowledge and perceptions about the Omicron variant. Comparison of the distribution of responses was evaluated for the following categories: (1) male vs. female, (2) doctors vs. other HCWs, (3) attended training/lectures vs. not attended, (4) participant from developed vs. developing countries according to the United Nations, using the Mann–Whitney U-test. Between-group comparison of knowledge and perception score variation across their health specialties was performed using analysis of covariance (ANCOVA). The covariates were age, sex, survey completion time, attended training/discussions regarding Omicron variant, and from developed and developing countries. The Chi-square test evaluated the differences between the subgroups’ overall knowledge and perception scores across the subgroups. Univariate and multivariate logistic regression analyses were performed to identify the factors associated with good knowledge and perception. Crude and adjusted odds ratios (OR) with 95% CI were calculated. Statistical significance was set as two-sided *p* ≤ 0.05.

### 2.7. Ethical Consideration

The study was conducted following the Checklist for Reporting Results of Internet E-Surveys (CHEERIES) guidelines [24] (Supplemental file: Appendix A). Ethical approval was obtained from the Ethical Review Committee of Ananthapuri Hospitals and Research Institute, India (AHRI/EC/Dec/2021). Participants were informed about the objective of the study, and their participation was completely voluntary, with no financial compensation. Electronic informed consent was shown on the initial page of the survey. Confidentiality of the information was assured throughout the study by making participants’ information anonymous and asking them to provide honest answers.

## 3. Results

### 3.1. Overview

A total of 940 of the 1054 HCWs that participated completed the survey questionnaire—a fulfillment rate of 89.1%, including 539 (57.3%) men and 401 (42.7%) women. Most participants were below 30 years of age (*n* = 530, 56.4%), and the mean age of the study cohort was 31.2 ± 11.2 years. Most participants were doctors (*n* = 430, 45.7%) or medical students (*n* = 225, 23.9%) and most were from Asia (*n* = 604, 64.3%). The participants’ average time to finish the survey was 5.9 ± 3.9 min. Participant characteristics are summarized in Table 1. All the HCW participants agreed that they had heard about the SARS-CoV-2 Omicron variant, but only 36.3% (*n* = 341) had the opportunity to attend lectures or discussions related to the Omicron variant.

### 3.2. Source of Information

When asked about the participants’ source of reliable information about the Omicron variant, the primary source mentioned was news media and government websites (Figure 1). About one-third of the participants reported that they often used news media (TV/video, newspapers, radio, and magazines), and a quarter (24.3%) relied on the websites of national and multinational health institutions (e.g., WHO, CDC, MOH, etc.) to obtain information on the Omicron variant. Furthermore, 39% of the participants sometimes held discussions on the Omicron variant with family and friends.

### 3.3. SARS-CoV-2 Variants of Concern

When the HCWs were asked to identify the origins of various WHO-designated SARS-CoV-2 VOCs, a high proportion identified the Omicron variant (B.1.1526) as having originated from South Africa but were not able to identify the origins of other SARS-CoV-2 VOCs correctly (see Figure 2 for more details).

### 3.4. Knowledge and Perceptions about SARS-CoV-2 Omicron Variant

We identified significant variance in the knowledge and perceptions between the HCWs and significant differences across HCWs when stratified by sex, doctors and other HCWs, developed and developing countries, and attended vs. did not attend lectures or discussions. Table 2 shows the level of HCWS knowledge and perceptions about the SARS-CoV-2 Omicron variant. Over 90% of the HCWs agreed that the Omicron variant could be transmitted to all age groups (91.4%), that following preventive measures (92.8%) can offer protection against the Omicron variant, and that taking two doses of a COVID-19 vaccine is important (89.1%). A higher proportion (80%) of doctors correctly identified that skin rash is not a symptom of Omicron compared to medical students (72.4%) and allied health professionals (69.5%).

Overall, only 20.1% (*n* = 194) of the HCWs exhibited good perceptions about the Omicron variant. Only 32.6% of the doctors believed that the currently available COVID-19 vaccines, and 40.2% that taking vaccine booster doses, offer protection against Omicron. Moreover, three-fourths of the doctors (25.8%) and allied health professionals (24.8%) believed that travel bans cannot control the global spread of Omicron, that the COVID-19 rapid test is not reliable for detecting Omicron (40%), and that steroids cannot treat severe Omicron (doctors: 46.5%); moreover, more than half of the HCWs perceived that COVID-19 mutations could alter the response to vaccines, treatment, and transmissibility. However, most of HCWs agreed that vaccinated and unvaccinated people are vulnerable to the Omicron variant. All the HCWs strongly agreed that countries should accelerate the COVID-29 vaccination program (92.6%).

### 3.5. Comparison of HCWs’ Knowledge and Perception Scores According to Their Health Specialties

Overall, doctors outperformed regarding knowledge and perceptions of the Omicron variant compared to other HCWs (Table 3). In the ANCOVA test, there was a statistically significant difference between the health specialties concerning the knowledge category (*p* = 0.004), with a median score of 14 points (0 to 20 points) for doctors and 12 points (4 to 20) for medical students, nurses, pharmacists, and allied health professionals. However, no significant difference in perception toward the SARS-CoV-2 Omicron variant was observed between the specialties (*p* = 0.827).

The difference is significant at *p* = <0.001. Covariates: age, sex, survey completion time, attended training/discussions regarding Omicron, and developed vs. developing countries.

### 3.6. Overall Knowledge and Perception Levels

The overall knowledge and perception scores for the SARS-CoV-2 Omicron variant statements across the subgroups are shown in Table 4. Based on modified Bloom’s criteria, only a quarter (24.3%) of the HCWs exhibited good knowledge about the Omicron variant, and a majority were male (14%) and from developing countries (18.9%). By contrary, only 20.6% of the HCWs demonstrated good perception. Moreover, good knowledge and perception were higher among doctors (13% and 10%) than other HCWs. A significant difference in knowledge levels was noted across all subgroups but not in perceptions.

### 3.7. Predictors of Good Knowledge and Perception

The multiple logistic regression indicated that HCWs who engaged in training/discussions concerning the SARS-CoV-2 Omicron variant were 1.80 times more likely to have good knowledge and perception than those HCWs who did not attend such trainings/discussions (Table 5). Furthermore, being older (OR: 1.01, 95% CI: 1.00–1.03), South American (OR: 2.69, 95% CI: 1.21–6.00), and Oceanic (OR: 2.04, 95% CI: 1.00–4.16) were significant predictors of higher perception scores. However, HCWs from Africa had significantly lower knowledge and perceptions.

## 4. Discussion

The SARS-CoV-2 Omicron variant is a hot topic of discussion among the public, especially among HCWs. This is the first study exploring HCWs’ understanding of the SARS-CoV-2 Omicron variant. The findings indicate that most health professionals have a comparable level of knowledge of this new variant, despite differences in knowledge and perception based on the region of practice. The Omicron variant has a high transmissibility rate—current figures indicate that it has reached ten times the number reported during the peak of the Delta wave [25]—creating an unprecedented burden to health officials and health systems. At present, the Omicron variant is causing a whirlwind worldwide, with thousands of cases reported every day, with no exemptions for COVID-19 vaccinated people [26]. To the authors’ knowledge, this study, for the first time, examined the basic professional knowledge and perception concerning the SARS-CoV-2 Omicron variant among HCWs.

The data from this international cross-sectional study indicate that HCWs are largely aware of the Omicron variant, but only a third (36.3%) of them had the opportunity to engage in lectures or discussions related to this new variant. We also found that a quarter of HCWs (24.3%) used official governmental websites as their go-to place as a primary source of information about Omicron. This study showed that HCWs reported having insufficient knowledge and some misperceptions, and there was a significant difference in knowledge levels across different categories of HCWs. This research also showed that 32% and 27% of the participants experienced poor knowledge and misperceptions towards the Omicron variant, respectively. Finally, this study showed that engaging in Omicron-related training/discussions was strongly associated with greater knowledge and improved perceptions of HCWs.

Knowledge about the Omicron variant varied significantly across the countries of the participants: those from developed countries had a higher overall score than their counterparts from non-developed countries. This difference could be for several reasons. For example, the lack of adequate infrastructure to enable adequate health responses to the pandemic, including a poor health communication strategy, could contribute to limited access to reliable information sources. In addition, other competing priorities in several developing settings such as socioeconomic crises that occur due to or are exacerbated by COVID-19 could divert the attention of communities, leading to complacency and poor uptake of health directions. Misinformation and disinformation are also contributing factors that amplify mistrust and impair health responses [27]—and this is likely to be more significant in settings with poor health communication strategies in place. Finally, COVID-19 fatigue or exhaustion due to the continually emerging variants and the need to adjust to these new realities is another potential contributing factor leading to complacency and poor response to emerging variants. However, this could apply to both developed and developing settings [28].

The poorer knowledge reported in developing settings could be, understandably, associated with lack of sufficient resources and means to sustainably fight this disease. The need for ongoing education and training of health professionals is resource-intensive, and developing settings may not be able to afford to provide them for a long time amidst ongoing competing health priorities. The World Health Organization (WHO) and other multilateral institutions have ongoing education, training, [29,30] and workshops on new developments in COVID-19 and available vaccines. However, national ministries and other relevant health authorities in individual countries have a significant responsibility to provide culturally and locally appropriate training and education to enable their health workforce to adequately tackle current and potential future variants. There is no doubt that the major challenge remains to address the significant inequity in vaccine distribution between the global south and north. However, improving the competency and preparedness of healthcare workers through appropriate training will be valuable in creating resilient health systems.

Due to the overwhelming number of cases of Omicron reported across the globe, many health systems are struggling to cope with the significant surge in demand for health needs. This must be seen in the context that health workers are highly likely to experience more severe COVID-19 cases [31]. Despite being characterized as “mild” in most settings, the Omicron variant still places considerable pressure on health systems [32], particularly making settings with low vaccination rates more exposed to the clinical and social consequences of the disease. Therefore, it is important to understand the differences between this particular variant and the previous ones and to clear some misperceptions related to the disease in an ongoing manner. Interestingly, our findings indicate that the probability of HCWs to identify VOCs and their reported places of origin has increased over time since the origin of COVID-19, potentially indicating an increased readiness of health professionals to become involved in knowing more about emerging variants. We also anticipate this to be the case with the wider public. Nevertheless, it is imperative to target HCWs through appropriate continuing education programs to equip them with the relevant knowledge to enable them to respond to the health crisis confidently and be able to educate vulnerable communities.

Generally, most participants had good knowledge about the individual measures to prevent and control the spreading of the SARS-CoV-2 Omicron variant and emphasized the importance of receiving two doses of the COVID-19 vaccine. However, wide discrepancies were identified in the perceptions of different categories of HCWs. For instance, 85% of the pharmacists and 74.2% of the doctors believed that the travel bans could limit the global spread of Omicron. Since the Omicron variant has already spread worldwide, travel bans go against WHO COVID-19 recommendation to apply a risk-based scientific approach when implementing specific travel measures and recommendations for reducing risk acquisition or spread of the virus [2]. Moreover, more than half of the doctors (53.5%) believed that steroidal therapy is effective against severe Omicron cases. The Omicron variant has numerous mutations in the spike protein (S-gene) [33], and drugs used in severe COVID-19 cases such as corticosteroids and interleukin-6 receptor blockers may not effectively target the spike protein in severe Omicron. Finally, a vast majority of HCWs strongly agreed that countries should accelerate the COVID-19 vaccination program (over 90%). Moreover, HCWs strongly recommended that older people and those with comorbidities should be advised to postpone travel to areas with community transmission.

### Strengths and Limitations

Responses obtained through online surveys are less reliable and accountable. However, given the pandemic situation, web-based design in medical research is growing. Our study is novel and it engaged HCWs from multiple countries to explore their knowledge and perceptions on SARS-CoV-2 Omicron. The large number of study participants (n = 940) and the high participation rate (89%) covering 101 countries across the six continents added significant strength. In addition, the survey tool was developed based on the WHO and CDC briefing materials for the detection, prevention, and control of the SARS-CoV-2 Omicron variant. The questionnaire developed was pilot tested, used close-ended questions to reduce information bias, and covered a wide range of Omicron-related topics, enabling us to explore the participants’ knowledge and perception multidimensionally. Moreover, we used the Qualtrics online software to obtain the responses within a 30 min window of time. The study collected specific demographic data to perform stratified comparison analyses.

However, this study has some limitations that should be considered. This research is a cross-sectional study conducted among HCWs during the second week of January 2022, when an alarming number of Omicron cases were being reported globally. The data presented in this study are self-reported and rely on participants’ honesty and recall ability. Thus, they may be subjected to recall bias. Moreover, the survey was conducted during the early stage of the Omicron epidemic, and thus, many of them may have had little or no knowledge about this new variant. Furthermore, factors such as internet access, variation in time, and language barriers may have influenced study participation. Despite these limitations, the findings from this study provide valuable information about HCWs’ knowledge and perceptions about Omicron during the peak stage of the pandemic.

## 5. Conclusions

This study found significant gaps in HCWs’ knowledge and significant misperceptions about the SARS-CoV-2 Omicron variant. As the world is experiencing a large wave of SARS-CoV-2 Omicron variant infections, which is inexorably reaching every continent, greater efforts through educational interventions targeting HCWs and the general population are urgently needed to improve their knowledge and perceptions of the SARS-CoV-2 Omicron variant.

## Figures and Tables

**Figure 1 healthcare-10-00438-f001:**
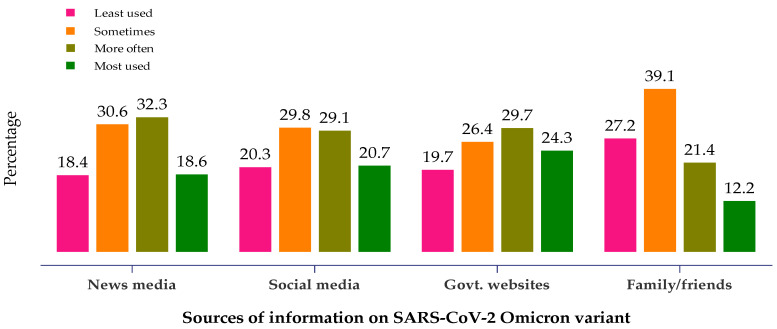
Healthcare workers’ sources of reliable information on the SARS-CoV-2 Omicron variant.

**Figure 2 healthcare-10-00438-f002:**
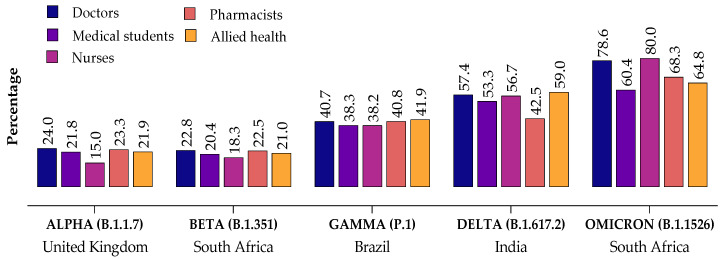
Healthcare workers correctly identified WHO-designated SARS-CoV-2 Variants of Concern (VOCs) and their countries of origin.

**Table 1 healthcare-10-00438-t001:** Characteristics of study participants (N = 940).

Variable	Frequency (%)
Sex	
Male	539 (57.3)
Female	401 (42.7)
Age (mean ± SD)	31.2 ± 11.2 years
18-29	530 (56.4)
30-39	194 (20.6)
≥40	216 (23)
Participant’s Location	
Asia	604 (64.3)
Africa	137 (14.6)
Europe	125 (13.3)
South America	39 (4.1)
North America	28 (3.0)
Oceanic	7 (0.7)
Profession	
Doctors	430 (45.7)
Medical students	225 (23.9)
Pharmacists	120 (12.8)
Nurses	60 (6.4)
Allied health	105 (11.2)
I Heard of SARS-CoV-2 Omicron	940 (100)
I Attended Lectures/Dissions About Omicron	341 (36.3)
Survey Completion Time (mean ± SD)	5.9 ± 3.9 min

SD: standard deviation.

**Table 2 healthcare-10-00438-t002:** Healthcare professionals’ knowledge and Perceptions about the SARS-CoV-2 Omicron.

Items Related to the SARS-CoV-2 Omicron Variant			Correct Response (%)							
Knowledge	*Doctors*	*Medical Students*	*Nurses*	*Pharmacists*	*Allied Health*	*Variance*	*p*-Values (Male vs. Female)	*p*-Value (Doctors vs. Others)	*p*-Value (Attended vs. Did Not Attend Lectures)	*p*-Value (Developed vs. Developing Country)
	*n* = 430	*n* = 225	*n* = 60	*n* = 120	*n* = 105					
The Omicron variant was first reported to the World health organization from South Africa	328 (76.3)	132 (58.7)	46 (76.7)	73 (60.8)	69 (65.7)	0.002	<0.001	0.001	0.010	<0.001
Skin rash is not a symptom of Omicron	341 (79.3)	163 (72.4)	45 (75)	83 (77.5)	73 (69.5)	0.315	0.767	0.048	0.952	0.060
Omicron can be transmited to all age groups	402 (93.5)	197 (87.6)	57 (95)	112 (93.3)	91 (86.7)	0.051	0.021	0.033	0.434	0.080
Wearing a face mask, hand hygiene, social distancing, indoor ventilation, avoiding crowded places, and COVID-19 vaccination can protect from Omicron	403 (93.5)	208 (92.4)	57 (95)	112 (93.3)	93 (88.6)	0.693	0.991	0.458	0.443	0.590
Taking two doses of COVID-19 vaccination is important	394 (91.6)	194 (86.2)	52 (86.7)	108 (90.0)	90 (85.7)	0.167	0.243	0.025	0.009	0.295
**Perceptions**										
Currently available COVID-19 vaccines offer protection against Omicron	140 (32.6)	74 (32.9)	17 (28.3)	41 (34.2)	34 (32.4)	0.888	0.177	0.978	0.855	0.326
A COVID-19 booster dose offers protection against Omicron	173 (40.2)	92 (40.9)	18 (30.0)	53 (44.2)	39 (37.1)	0.331	0.149	0.606	0.269	0.906
COVID-19 mutations could alter the response to vaccines, treatment, and transmissibility	264 (61.4)	112 (49.8)	38 (63.3)	67 (55.8)	60 (57.1)	0.044	0.516	0.030	0.244	0.954
Both vaccinated and unvaccinated people are vulnerable to Omicron	388 (90.2)	196 (87.1)	56 (93.3)	104 (86.7)	89 (84.8)	0.294	0.335	0.152	0.125	0.989
Travel bans cannot limit the global spread of Omicron	111 (25.8)	48 (21.3)	18 (30)	18 (15)	26 (24.8)	0.085	0.004	0.126	0.324	<0.001
Older people and people with comorbidities should postpone travel	396 (92.1)	211 (93.8)	54 (90)	109 (90.8)	95 (90.5)	0.764	0.807	0.941	0.193	0.314
Steroids are not effective against severe Omicron	200 (46.5)	114 (50.7)	33 (55)	69 (57.5)	72 (68.6)	<0.001	0.009	0.002	0.413	0.678
A COVID-19 rapid antigen test is not reliable to detect the Omicron	174 (40.5)	100 (44.4)	25 (41.7)	54 (45)	44 (41.9)	0.847	0.475	0.314	0.998	0.060
Face masks offer protection against all SARS-CoV-2 variants	370 (86)	191 (84.9)	46 (76.7)	104 (86.7)	82 (78.1)	0.122	0.501	0.192	0.419	0.125
Countries should accelerate the COVID-19 vaccination program	405 (94.2)	206 (91.6)	54 (90)	109 (90.8)	96 (91.4)	0.516	0.472	0.080	0.381	0.179

Variance: ANOVA test (*p*-value): all the comparisons were analyzed using a Mann–Whitney U-test.

**Table 3 healthcare-10-00438-t003:** ANCOVA test to compare the participants’ knowledge and perception score variations across health specialties.

	Specialty	Frequency	Median (Range)	F	df	*p*-Value
**Knowledge score**	Doctors	430	14 (0–20)	3.82	4	0.004
	Medical students	225	12 (4–20)			
	Nurses	60	12 (6–20)			
	Pharmacists	120	12 (2–20)			
	Allied Health	105	12 (2–20)			
**Perception score**	Doctors	430	13 (5–18)	0.374	4	0.827
	Medical students	225	12 (6–20)			
	Nurses	60	12 (6–18)			
	Pharmacists	120	12 (6–20)			
	Allied Health	105	12 (4–18)			

**Table 4 healthcare-10-00438-t004:** Overall knowledge and perception levels across subgroups.

Subgroup		Knowledge			*p*-Value	Perception			*p*-Value
		Good	Moderate	Poor		Good	Neutral	Misperceptions	
	Scores	16–20	12–15	<12		16–20	12–15	<12	
Total	N= 940	228 (24.3)	409 (43.5)	303 (32.2)	<0.001	194 (20.6)	492 (52.3)	254 (27)	0.001
Sex					0.624				0.913
	Male	132 (14)	240 (25.5)	167 (17.8)		109 (11.6)	285 (30.3)	145 (15.4)	
	Female	96 (10.2)	169 (18)	136 (14.5)		85 (9)	207 (22)	109 (11.6)	
Age (years)					0.002				0.132
	<30	118 (12.6)	216 (23)	196 (20.9)		97 (10.3)	286 (30.4)	147 (15.6)	
	≥30	110 (11.7)	193 (20.5)	107 (11.4)		97 (10.3)	206 (21.9)	107 (11.4)	
Countries					<0.001				0.369
	Developing	178 (18.9)	312 (33.3)	266 (28.3)		43 (4.6)	98 (10.4)	43 (4.6)	
	Developed	50 (5.3)	97 (10.3)	37 (3.9)		151 (16.1)	394 (41.9)	211 (22.4)	
Profession					0.001				0.496
	Doctors	121 (12.9)	203 (21.6)	106 (11.3)		94 (10)	227 (24.1)	109 (11.6)	
	Medical students	42 (4.5)	87 (9.3)	96 (10.2)		46 (4.9)	109 (11.6)	70 (7.4)	
	Nurses	15 (1.6)	27 (2.9)	18 (1.9)		12 (1.3)	36 (3.8)	12 (1.3)	
	Pharmacists	27 (2.9)	50 (5.3)	43 (4.6)		18 (1.9)	68 (7.2)	32 (3.6)	
	Allied health	23 (2.4)	42 (4.5)	40 (4.3)		24 (2.6)	52 (5.5)	29 (3.1)	

**Table 5 healthcare-10-00438-t005:** Factors associated with good knowledge and perception about the SARS-CoV-2 Omicron.

	Odds Ratio (95% Confidence Intervals)			
	Categories	Good Knowledge	Good Perception	Both
Sex	Female	1	1	1
	Male	0.93 (0.68–1.27)	0.89 (0.64–1.24)	0.85 (0.49–1.47)
Age	Years	1.07 (0.99–1.02)	1.01 (1.00–1.03) *	1.01 (0.99–1.04)
Location	Asia	1	1	1
	Africa	0.48 (0.26–78) **	0.55 (0.31–0.95) *	0.21 (0.05–0.92) *
	Europe	1.18 (0.76–1.84)	1.14 (0.70–1.84)	1.33 (0.66–2.68)
	South America	1.23 (0.53–2.86)	2.69 (1.21–6.00) *	1.37 (0.37–5.00)
	North America	0.53 (0.06–4.52)	2.84 (0.62–12.91)	-
	Oceanic	0.73 (0.33–1.63)	2.04 (1.00–4.16) *	0.90 (0.25–3.22)
Profession	Doctors	1	1	1
	Medical students	0.67 (0.43–1.04)	1.15 (0.70–1.76)	0.85 (0.37–1.94)
	Pharmacists	0.91 (0.48–1.72)	0.86 (0.43–1.70)	1.08 (0.40–2.96)
	Nurses	0.85 (0.51–1.42)	0.72 (0.40–1.30)	0.59 (0.19–1.81)
	Allied health	0.77 (0.46–1.29)	1.07 (0.64–1.80)	0.78 (0.31–1.95)
Engaged in Omicron-related training/discussions	No	1	1	1
	Yes	1.72 (1.27–2.38) **	0.98 (0.70–1.38)	1.80 (1.04–3.11) **

*Adjusted:* age, gender, location, profession, and engaged in training/discussion. * *p* < 0.05; ** *p* < 0.01.

## Data Availability

All the data presented in this study are available upon request from the corresponding author.

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
