# Peer review of "Healthcare Workers’ Knowledge and Perception of the SARS-CoV-2 Omicron Variant: A Multinational Cross-Sectional Study"

_healthcare, 2022, doi:10.3390/healthcare10030438_

Round 1

Reviewer 1 Report

This study is well written and properly scientifically structured. However, the authors have to address all comments/concerns to ensure that all problems affecting the sobriety of the research are fixed.

  • Introduction Section: The authors did not explain the shortcomings of previous studies of the same topic that led to this study. How does this study differ from previous studies? It should be explained in detail to highlight the novelty of this study.
  • Figures and Tables: All figures are drawn in high resolution. Table 1 requires alignment of contents in columns
  • English Writing: This article is written in good English. However, this paper requires minor proofreading of the entirety of the paper to eliminate all the problems associated with typos, spelling, and grammar mistakes.
  • List of references: The references are recent and related to the paper topic. However, it should follow accurately the MDPI-Healthcare style. Also, the number of references for this study is insufficient. Some links in the reference list are not working such as references [1], [2], [6], [10] … etc. This paper requires moderate check of the reference list.

Author Response

This study is well written and properly scientifically structured. However, the authors have to address all comments/concerns to ensure that all problems affecting the sobriety of the research are fixed.

Reviewer comments: Introduction Section: The authors did not explain the shortcomings of previous studies of the same topic that led to this study. How does this study differ from previous studies? It should be explained in detail to highlight the novelty of this study.

Author response: We appreciate your suggestions and added these details in the Introduction section Line 243 - 248. 

Reviewer comments: Figures and Tables: All figures are drawn in high resolution. Table 1 requires alignment of contents in columns. 

Author response: All the images are drawn by PRISMA Graphpad. The images are high-resolution and the pixels can be adjusted based on journal requirements. Table 1 was assigned following your suggestions. 

Reviewer comment: English Writing: This article is written in good English. However, this paper requires minor proofreading of the entirety of the paper to eliminate all the problems associated with typos, spelling, and grammar mistakes.

List of references: The references are recent and related to the paper topic. However, it should follow accurately the MDPI-Healthcare style. Also, the number of references for this study is insufficient. Some links in the reference list are not working such as references [1], [2], [6], [10] … etc. This paper requires moderate check of the reference list.

Authors response: We have edited the English where required and we also included additional references to substantiate our paper. All the references were cited according to the Healthcare Journal reference guidelines. 

Reviewer 2 Report

In general, I think that the introduction and the discussion can improve if they take up more of their results, no emphasis is placed on the results themselves to discuss, since only they generalize about them and on some occasions if they point out a specific result. Table 2 could be improved if the percentage of those who knew the information is presented with respect to those who did not, so that the result is clearer (% yes vs % no,  for example).

Author Response

Reviewer Comments: In general, I think that the introduction and the discussion can improve if they take up more of their results, no emphasis is placed on the results themselves to discuss, since only they generalize about them and on some occasions if they point out a specific result. Table 2 could be improved if the percentage of those who knew the information is presented with respect to those who did not, so that the result is clearer (% yes vs % no,  for example).

Author response: We improved the introduction and discussion sections and provided more information on previous studies in the introduction (Line 236 - 241) and interpreted results in the discussion section (Line 678 - 696). Table 2 was slightly modified by adding the number of health professionals in each section. We provided only correct responses in Table 2. 

Reviewer 3 Report

In this excellent manuscript, the authors compare the level of awareness of the SAARS-CoV-2 Omicron variants in various populations of HCW around the time that it was just emerging.  It makes the valid point that there was a  scarcity of information about the variant and, indeed, some disinformation.  One caveat that I found lacking though was the impact of “Covid exhaustion”, i.e.  the fact that people are genuinely tired of the roller coaster that has been Covid.  Once we feel we are turning the corner, a new variant emerges and we have to adjust our strategy.  This caveat should be mentioned in the manuscript.

Author Response

Reviewer comments: In this excellent manuscript, the authors compare the level of awareness of the SAARS-CoV-2 Omicron variants in various populations of HCW around the time that it was just emerging.  It makes the valid point that there was a  scarcity of information about the variant and, indeed, some disinformation.  One caveat that I found lacking though was the impact of “Covid exhaustion”, i.e.  the fact that people are genuinely tired of the roller coaster that has been Covid.  Once we feel we are turning the corner, a new variant emerges and we have to adjust our strategy.  This caveat should be mentioned in the manuscript.

Author response: Thank you for this suggestion. This has now been included on page 9 from line 289 - 293.

Reviewer 4 Report

This is very important and timely research since the effectiveness of their treatment of patients largely depends on how deep and correct the knowledge of medical workers about Omicron is. This is a first study that provides valuable and reasonable information that is important for the understanding of the necessity of improving the knowledge of Omicron hazard perception by HCWs. The MS is clear and well–written, however, some modifications are required.

Point 1: Abstract. I am not sure that it is necessary to number subsections of abstracts.

Point 2: Line 46–47. I would recommend updating information on total cases and total deaths from COVID–19 and mentioning the exact day when information has been obtained. For instance, “According to the Johns Hopkins University, as of DD/MM/YY, global COVID–19 activity was…” in addition, please, update all other references taken from websites of WHO and CDC with this regard. Information is rapidly outdated.

Point 3: Line 53–55. The authors said that “The Omicron variant remains largely less severe than earlier Variants of Concern (VOC), and the significantly high number of vaccinated populations in certain settings potentially reduces morbidity and mortality.” Please, explain the second part of this sentence (“the significantly high number of vaccinated populations in certain settings potentially reduces morbidity and mortality”). Do you mean here something very general or are you talking about Omicron? In the last case it does not match with the statement on Lines 22–23 (“…currently available COVID–22 vaccines showed little or no protection against the Omicron variant…”) or on Lines 59–61 (“…the existing vaccines provide the greatest protection against the Delta variant and provide some level of protection against Omicron…”).

Point 4: Table 1. How can the authors explain the fact that the majority of respondents were from Asia (64.3%), while respondents from Europe or America were only 3–13%? Does this mean that interest in the topic is significantly higher in Asia?

Point 5: Table 1 is hard to read due to poor layout. Please, consider capitalizing on the main points (SEX, LOCATION, PROFESSION). What do the authors mean under "Allied health"? Is it "Allied health professions"? Two lines ("I heard of SARS–CoV–2” and “I attended lectures/discussions about Omicron" should have their subtitle like "HOW INFORMATION REGARDING OMICRON WAS OBTAINED?" "SURVEY COMPLETION TIME" should become a separate point and be capitalized.

Point 6: Line 353. Please, replace “…this study provide…” with “…this study provides…”

Point 7: Line 354. Please, add a dot at the end of the sentence.

Point 8: Line 101. The authors refer to Appendix 1. Please, refer to Appendix 1 correctly: Figure A1, or Table A1.

Point 9: Line 160. The authors refer to Appendix 2. Please, refer to Appendix 2 correctly: Figure A2, or Table A2.

Point 10: Line 258–259. Please, confirm this statement with relevant references.

Point 11: Line 256–267. The sentence “At present, the Omicron variant is a whirlwind worldwide, with thousands of cases reported every day, with no exemptions to HCWs and COVID–19 vaccinated people” should be rephrased, because HCWs are also (i) people and (ii) most of them are vaccinated. I suggest removing HCWs from this sentence.

Point 12: Line 362. Please, provide information on Appendixes in the manner described in the Microsoft Word template with proper titles and proper designations.

Point 13: Line 363. The authors stated that they provided three Appendixes. Please find an appropriate place in the text for referring to Appendix 3. The reference to it is missing.

Point 14: It seems to me that the archive of supplementary files is broken - it cannot be unzipped.

Author Response

Reviewer: This is very important and timely research since the effectiveness of their treatment of patients largely depends on how deep and correct the knowledge of medical workers about Omicron is. This is a first study that provides valuable and reasonable information that is important for the understanding of the necessity of improving the knowledge of Omicron hazard perception by HCWs. The MS is clear and well–written, however, some modifications are required.

Authors response: We appreciate your timely review and constructive comments and suggestions. We modified the manuscript following your suggestions and the responses to your points are provided below. 

Point 1: Abstract. I am not sure that it is necessary to number subsections of abstracts.

Authors response: We removed the subsections in the abstract. 

Point 2: Line 46–47. I would recommend updating information on total cases and total deaths from COVID–19 and mentioning the exact day when information has been obtained. For instance, “According to the Johns Hopkins University, as of DD/MM/YY, global COVID–19 activity was…” in addition, please, update all other references taken from websites of WHO and CDC with this regard. Information is rapidly outdated.

Authors response: We updated the information in the introduction section and updated the references. 

Point 3: Line 53–55. The authors said that “The Omicron variant remains largely less severe than earlier Variants of Concern (VOC), and the significantly high number of vaccinated populations in certain settings potentially reduces morbidity and mortality.” Please, explain the second part of this sentence (“the significantly high number of vaccinated populations in certain settings potentially reduces morbidity and mortality”). Do you mean here something very general or are you talking about Omicron? In the last case it does not match with the statement on Lines 22–23 (“…currently available COVID–22 vaccines showed little or no protection against the Omicron variant…”) or on Lines 59–61 (“…the existing vaccines provide the greatest protection against the Delta variant and provide some level of protection against Omicron…”).

Response: Thank you for pointing this out. We have now amended the document to make sure there is consistency throughout and also in light of emerging evidence in this area. 

Point 4: Table 1. How can the authors explain the fact that the majority of respondents were from Asia (64.3%), while respondents from Europe or America were only 3–13%? Does this mean that interest in the topic is significantly higher in Asia?

Author response: We distributed responses in various social sites thus, we are not sure about the differences in the interest based on geographic differences. 

Point 5: Table 1 is hard to read due to poor layout. Please, consider capitalizing on the main points (SEX, LOCATION, PROFESSION). What do the authors mean under "Allied health"? Is it "Allied health professions"? Two lines ("I heard of SARS–CoV–2” and “I attended lectures/discussions about Omicron" should have their subtitle like "HOW INFORMATION REGARDING OMICRON WAS OBTAINED?" "SURVEY COMPLETION TIME" should become a separate point and be capitalized.

Authors response: We presented this subheading in the UPPERCASE. 

Point 6: Line 353. Please, replace “…this study provide…” with “…this study provides…”

Authors response: We modified the sentence 

Point 7: Line 354. Please, add a dot at the end of the sentence.

Authors response: Added 

Point 8: Line 101. The authors refer to Appendix 1. Please, refer to Appendix 1 correctly: Figure A1, or Table A1.

Author response: We added the references of the Appendix correctly. 

Point 9: Line 160. The authors refer to Appendix 2. Please, refer to Appendix 2 correctly: Figure A2, or Table A2.

Authors response: Amended 

Point 10: Line 258–259. Please, confirm this statement with relevant references.

Authors response: Added the references to this sentence. 

Point 11: Line 256–267. The sentence “At present, the Omicron variant is a whirlwind worldwide, with thousands of cases reported every day, with no exemptions to HCWs and COVID–19 vaccinated people” should be rephrased, because HCWs are also (i) people and (ii) most of them are vaccinated. I suggest removing HCWs from this sentence.

Authors response: Rephrased the sentence. 

Point 12: Line 362. Please, provide information on Appendixes in the manner described in the Microsoft Word template with proper titles and proper designations.

Authors response: We added these details in the Appendix. 

Point 13: Line 363. The authors stated that they provided three Appendixes. Please find an appropriate place in the text for referring to Appendix 3. The reference to it is missing.

Authors response: We added these details in the appendix section. 

Point 14: It seems to me that the archive of supplementary files is broken - it cannot be unzipped.

Authors response: We uploaded the supplementary files in the figshare and provided the link to download the zip files directly from the open-source. https://doi.org/10.6084/m9.figshare.19170032 

This manuscript is a resubmission of an earlier submission. The following is a list of the peer review reports and author responses from that submission.